

# An enhanced BERT model with improved local feature extraction and long-range dependency capture in promoter prediction for hearing loss

Jing Sun[1], Yangfan Huang[1], Jiale Fu[1], Li Teng[1], Xiao Liu[1] and Xiaohua Luo[2]

[1] School of Microelectronics and Communication Engineering, Chongqing University, Chongqing, China
[2] Department of Otolaryngology Surgery, Chongqing University FuLing Hospital, Chongqing, China

Corresponding authors
Xiao Liu, liuxiao@cqu.edu.cn
Xiaohua Luo, 354206545@qq.com

## ABSTRACT

Promoter prediction has a key role in helping to understand gene regulation and in developing gene therapies for complex diseases such as hearing loss (HL). While traditional Bidirectional Encoder Representations from Transformers (BERT) models excel in capturing contextual information, they often have limitations in simultaneously extracting local sequence features and long-range dependencies inherent in genomic data. To address this challenge, we propose DNABERT-CBL (DNABERT-2_CNN_BiLSTM), an enhanced BERT-based architecture that fuses a convolutional neural network (CNN) and a bidirectional long and short-term memory (BiLSTM) layer. The CNN module is able to capture local regulatory features, while the BiLSTM module can effectively model long-distance dependencies, enabling efficient integration of global and local features of promoter sequences. The models are optimized using three strategies: individual learning, cross-disease training and global training, and the performance of each module is verified by constructing comparison models with different combinations. The experimental results show that DNABERT-CBL outperforms the baseline DNABERT-2_BASE model in hearing loss promoter prediction, with a 20% reduction in loss, a 3.3% improvement in the area under the working characteristic curve (AUC) of the subjects, and a 5.8% improvement in accuracy at a sequence length of 600 base pairs. In addition, DNABERT-CBL consistently outperforms other state-of-the-art BERT-based genome models on several evaluation metrics, highlighting its superior generalization ability. Overall, DNABERT-CBL provides an effective framework for accurate promoter prediction, offers valuable insights into gene regulatory mechanisms, and supports the development of gene therapies for hearing loss and related diseases.

# INTRODUCTION

Promoters are non-coding sequence regions located near transcription start sites (TSS) in genomic DNA. They serve as initiation points for gene transcription and play a critical role

in gene regulation and expression. Research has established a strong association between promoters and complex diseases such as diabetes (*Döhr et al., 2005*; *Ionescu-Tîrgovişte, Gagniuc & Guja, 2015*), cancer (*Davuluri et al., 2008*), and Huntington's disease (*Coles, Caswell & Rubinsztein, 1998*). In recent studies, the significance of promoter mutations in the onset of hearing loss has also been revealed. For instance, mutations in the core promoter of the GJB2 gene are a major cause of non-syndromic recessive hearing loss (*Matos et al., 2007*), and whole-genome sequencing has underscored the close link between GJB2 promoter mutations and hearing loss (*Le Nabec et al., 2021*). Furthermore, variants in the SOD2 promoter are considered key contributors to age-related hearing loss (ARHL) (*Nolan et al., 2013*), and the role of promoter DNA methylation in hearing loss has been further validated (*Xu et al., 2020*). Variations in promoter sequences not only influence disease occurrence but may also serve as targets for gene therapy. For instance, variations in the MYO7A promoter affect gene expression and may act as potential modifiers for DFNA11 hearing loss (*Street et al., 2011*). Studies have shown that promoter-driven gene therapy, such as Myo15 promoter-mediated hair cell-specific gene therapy, holds potential clinical applications for various forms of hearing impairment, including autosomal recessive deafness (*Wang et al., 2024a*), thus underscoring the importance of promoters as therapeutic targets in hearing loss research (*Aaron et al., 2023*).

In recent years, gene sequences have increasingly been conceptualized as a "molecular language" essential for the regulation of gene expression. Analogous to natural language, gene sequences exhibit complex patterns and contextual dependencies. Advances in natural language processing (NLP), particularly in large language models such as BERT (Bidirectional Encoder Representations from Transformers) (*Devlin et al., 2019*), have provided new insights into the analysis of genetic sequences. Due to its bidirectional encoding, BERT can effectively capture contextual information within sequences. This architecture has demonstrated substantial potential in NLP applications and holds promise for genomic research. However, genetic data possess unique characteristics, including the presence of local regulatory sites and long-range regulatory patterns, which pose challenges for applying BERT to gene sequence analysis effectively (*Choi & Lee, 2023*; *Benegas et al., 2024*).

Specifically, gene sequences frequently contain abundant local regulatory sites and short-range patterns (*Solovyev, Shahmuradov & Salamov, 2010*), necessitating models with robust local feature extraction capabilities. Furthermore, gene regulation is also influenced by distal regulatory elements within the three-dimensional (3D) chromatin conformation. For example, distal elements can regulate gene expression through chromatin loops that bring them into contact with promoters (*Holwerda & De Laat, 2012*). Thus, promoter prediction tasks require models that not only capture local regulatory information but also effectively model complex long-range dependencies, enabling the identification of promoter interactions with distal regulatory elements. Relying solely on BERT's self-attention mechanism may therefore be insufficient to fully capture these intricate regulatory features, highlighting the need for complementary models that can enhance both local and global feature extraction.

To address these challenges, recent studies have explored the integration of BERT with other deep learning models to enhance predictive performance. Convolutional neural networks (CNNs) are well-known for their strong local feature extraction capabilities, utilizing sliding windows to effectively capture local patterns within sequences. When combined with BERT, CNNs significantly improve the model's ability to process local features. For example, *Kaur & Kaur (2023)* demonstrated substantial improvements in demand classification by incorporating convolutional modules into the BERT framework, while *Chen, Cong & Lv (2022)* successfully integrated attention mechanisms with CNN-based local feature extraction modules, thereby improving the accuracy of long-text classification.

In contrast, long short-term memory (LSTM) networks excel at modeling long-range dependencies, addressing some of BERT's limitations in capturing information from longer sequences. Integrating BERT with LSTM networks further enhances the model's ability to handle complex long-distance dependencies. For instance, *Talaat (2023)* proposed a hybrid sentiment analysis system combining BERT with bidirectional LSTM (BiLSTM) and bidirectional gated recurrent units (BiGRU), achieving notable improvements in classification accuracy. Similarly, *Cao, Zhang & Huang (2024)* developed a hybrid architecture for multi-task real-time prediction by leveraging the strengths of both LSTM and Transformer models.

In the field of bioinformatics, the integration of BERT with CNNs or LSTMs has demonstrated promising results. For example, the BERT-TFBS model (*Wang et al., 2024b*) utilizes BERT to extract global contextual features from DNA sequences, which are then refined by CNNs to capture local patterns relevant to transcription factor binding site prediction. Furthermore, *Bokharaeian, Dehghani & Diaz (2023)* proposed a model combining PubMedBERT with LSTM to extract SNP-phenotype associations from biomedical texts, showing strong performance in high-confidence association extraction. More recently, the iProL model (*Peng, Sun & Fan, 2024*) employed a Longformer-based architecture with CNN and Bi-LSTM modules to identify DNA promoters from raw sequences, highlighting the value of integrating pre-trained language models with both local and sequential feature extractors. Collectively, these efforts underscore the utility of hybrid models in advancing genomic prediction tasks.

Building on these findings, we propose the DNABERT-CBL (DNABERT-2_CNN_BiLSTM) model, which integrates BERT, CNN, and BiLSTM components to enhance model performance. This architecture leverages BERT's strength in capturing contextual dependencies, CNN's robust local feature extraction capabilities, and BiLSTM's ability to model bidirectional contextual information, thereby improving the model's capacity to capture both short-range and long-range dependencies. By balancing global and local feature extraction, DNABERT-CBL significantly enhances gene sequence prediction performance. Through this multi-component integration, the model effectively identifies key regulatory sites within promoters and captures global regulatory relationships within promoter sequences. Specifically, CNN modules focus on extracting short-range dependencies and local patterns, while BiLSTM, as a bidirectional extension of LSTM, processes sequence information in both forward and backward directions, thereby

enriching the model's ability to model long-range dependencies and global sequence structures.

To evaluate the effectiveness of the DNABERT-CBL model, we utilized data from six databases encompassing 1,099 genes associated with three diseases: hearing loss, breast cancer, and cervical cancer. Promoter sequences were extracted from the EPDnew database, while non-promoter sequences were obtained from the NCBI database. We designed three fine-tuning strategies: (1) individual training, (2) cross-disease training, and (3) global training. Experimental results demonstrate that DNABERT-CBL consistently outperforms the baseline model (DNABERT-2_BASE) in both accuracy and stability. Under a sequence length of 600 base pairs, loss values decreased by 20 percentage points, while area under the curve (AUC) and accuracy (ACC) improved by 3.3% and 5.8%, respectively. Furthermore, compared to three other advanced BERT-based genomic models, DNABERT-CBL consistently demonstrated significant performance improvements across all evaluation metrics, thus confirming its robustness and generalizability.

## MATERIALS AND METHODS

### Data sources

In this study, we utilized datasets corresponding to three specific diseases: (1) Hearing loss, (2) breast cancer, and (3) cervical cancer. Hearing loss-related gene data were obtained from the following four databases: ClinVar (*Landrum et al., 2020*), DVD (*Azaiez et al., 2018*) (Deafness Variation Database), OMIM (https://www.omim.org), and HHL (*Bolz, 2016*) (Hereditary Hearing Loss). Breast cancer data were sourced from METABRIC (*Mukherjee et al., 2018*) (Molecular Taxonomy of Breast Cancer International Consortium), and cervical cancer data were obtained from CCDB (*Agarwal et al., 2011*) (Cancer of the Cervix Database). Gene entries from each database were extracted, deduplicated, and summarized; the number of genes sourced from each database is presented in Table 1.

Experimentally validated human promoter sequences were obtained from the EPDnew database. Based on the disease-related genes identified above, we subsequently matched the corresponding promoter sequences in EPDnew to construct the positive dataset. For the negative dataset, we first identified all experimentally validated human promoters within the genome and removed them, resulting in a genome-wide sequence dataset devoid of promoters (*Umarov et al., 2019*). From this dataset, we randomly selected contiguous genomic sequences matching the lengths of actual biological promoters, thereby generating the negative samples.

To assess the impact of sequence length on model performance, we considered two promoter sequence lengths: 300 base pairs (bp) (*Wang et al., 2023*) (covering positions −249 to +50 relative to the transcription start site, TSS) and 600 bp (*Dogan et al., 2015*) (spanning positions −499 to +100 relative to the TSS). Here, "+50" and "+100" indicate the number of base pairs downstream of the TSS. This approach acknowledges that promoter regions encompass not only upstream regulatory elements but also downstream regions that are critical for transcriptional regulation.

**Table 1 Disease databases and gene data counts.** Summary of the disease databases and their corresponding gene data counts for various diseases. It includes data sources like ClinVar, METABRIC, and CCDB, with the total number of related genes and total gene count for each dataset.

| Disease | Hearing loss | | | | Breast cancer | Cervical cancer |
|---|---|---|---|---|---|---|
| Data source | ClinVar | DVD | OMIM | HHL | METABRIC | CCDB |
| Related gene count | 376 | 223 | 154 | 122 | 173 | 538 |
| Total gene count | 388 | | | | 173 | 538 |

To evaluate the influence of different datasets on model prediction performance, we designed three fine-tuning strategies:

(1) Individual learning: Both model training and testing were conducted exclusively on the hearing loss dataset.

(2) Cross-disease training: Model training was performed using publicly available promoter datasets from breast cancer and cervical cancer, with testing conducted on the hearing loss dataset. This approach aims to leverage larger datasets from related diseases to enhance model performance when the hearing loss dataset is limited in size. Additionally, it allows an assessment of the model's transferability and generalization across different disease contexts.

(3) Global training: Model training incorporated all known human promoter datasets, with testing performed on the hearing loss dataset. This strategy maximizes the available training data to achieve robust model learning.

The data distribution for training, validation, and testing under these three strategies is detailed in Table 2. Collectively, these experimental strategies facilitate a systematic evaluation of how disease-specific and cross-disease datasets affect the model's ability to predict hearing-loss-related promoters, providing insights into its generalizability and robustness.

## DNABERT-2 model

DNABERT-2 (*Zhou et al., 2024*) is an enhanced model built upon the widely used Transformer architecture (*Vaswani et al., 2017*), specifically developed for DNA sequence analysis. Building on the foundation of its predecessor DNABERT (*Ji et al., 2021*), which demonstrated notable performance in genomic tasks, DNABERT-2 introduces several key improvements. It incorporates byte pair encoding (BPE) (as illustrated in Fig. 1) as a replacement for the traditional k-mer method, thereby enhancing computational efficiency and enabling better sequence compression. Furthermore, Attention with Linear Biases (ALiBi) replaces the conventional positional embeddings, thus improving the model's capability to handle longer sequences more effectively. Moreover, the integration of Flash Attention and the GeGLU (gated linear unit (GLU) and generalized linear unit (GELU)) activation function further enhances the model's performance and computational resource efficiency. Collectively, these advancements enable DNABERT-2 to achieve superior accuracy and efficiency in a variety of genomic analysis tasks.

**Table 2 Data distribution for three fine-tuning strategies.** The distribution of data for three different fine-tuning strategies: individual learning, cross-disease training, and global training. The table breaks down the data into training, validation, and test sets for each strategy. It highlights the varying dataset sizes, with individual learning utilizing a smaller dataset, and global training involving a significantly larger dataset for more robust model performance.

| Tactics | All | Train | Validation | Test |
|---|---|---|---|---|
| Individual learning | 1,194 | 716 | 238 | 240 |
| Cross-disease training | 1,830 | 508 | 128 | 1,194 |
| | 3,008 | 1,450 | 364 | 1,194 |
| Global training | 59,195 | 46,401 | 11,600 | 1,194 |

| **Byte Pair Encoding** |
|---|

| Iteration | Training corpus | Vocabulary |
|---|---|---|
| 0 | A T G C G A T A G C T A G C | [A,G,C,T] |
| 1 | A T G C G A **TA** G C **TA** G C | [A,G,C,T,**TA**] |
| 2 | A T G C G A **TA** **GC** **TA** **GC** | [A,G,C,T,**TA**,**GC**] |
| 3 | A T G C G A **TA** **GC** **TA** **GC** | ...... |

**Figure 1 Byte pair encoding.** The process of byte pair encoding (BPE) applied to genomic sequences. In each iteration, the most frequent pair of symbols is merged, thereby expanding the vocabulary and reducing the sequence length. The evolution of the training *corpus* is shown across different iterations, where specific nucleotide pairs (like "TA" and "GC") are progressively merged, updating the vocabulary with new entries.

## Feature extraction strategy *via* multi-model integration

In genomic sequence analysis, extracting comprehensive features from DNA sequences is essential to capture both global context and local sequence patterns. To address this challenge, we propose a multi-model integration strategy that leverages the strengths of different deep learning architectures.

BERT, based on a bidirectional encoder architecture, reads sequences from both left to right and right to left, making it highly effective for capturing contextual information. Moreover, BERT's extensive pre-training on large-scale unlabeled data enables it to learn universal sequence representations, which can then be fine-tuned for various downstream tasks.

However, gene sequences typically contain a substantial number of local regulatory sites and short-range patterns that require specialized local feature analysis. CNNs, with their local convolutional operations, efficiently extract such features from sequences. Consequently, we initially combined BERT with CNNs so that CNNs could refine the local regions of the global features extracted by BERT. Through convolutional kernels and pooling operations, CNN is effective in capturing micro-patterns and local structures, which is crucial for identifying short-range dependencies within promoter sequences.

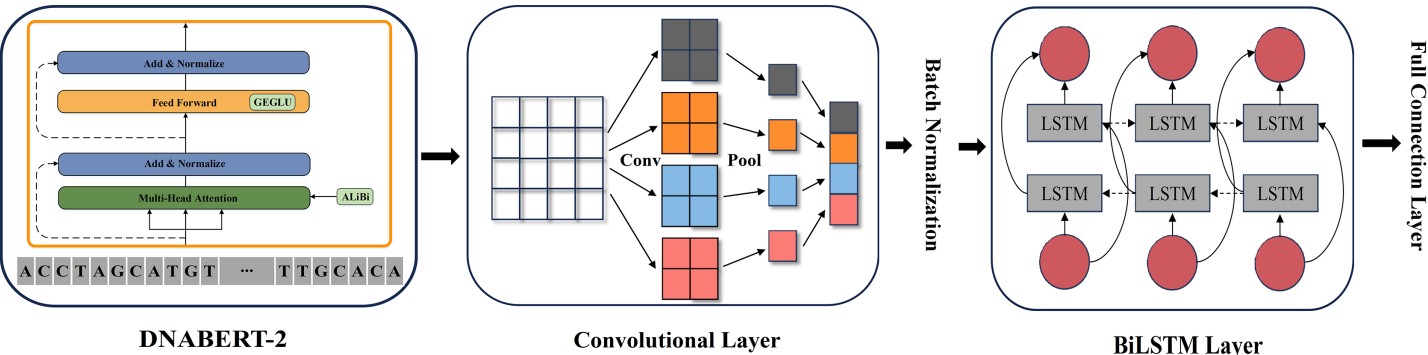

**Figure 2 DNABERT-2_CNN_BiLSTM architecture.** The architecture of DNABERT-2_CNN_BiLSTM, which is a hybrid deep learning model combining convolutional neural networks (CNN) and bidirectional long short-term memory (BiLSTM) layers. It processes genome sequences, performing feature extraction through convolution, followed by sequence encoding using LSTM layers, ultimately leading to predictions based on learned patterns in the data.                               

Furthermore, complex long-range dependencies play a critical role in gene regulation, especially through three-dimensional chromatin conformations. In this process, distal regulatory elements interact with promoters *via* chromatin loops, thereby influencing gene expression. To address these dependencies, we integrated BiLSTM, which has a bidirectional architecture that simultaneously captures both forward and backward dependencies, enabling comprehensive modeling of long-range relationships, particularly in recognizing distant regulatory elements.

As illustrated in Fig. 2, we constructed a multi-layer feature extraction architecture based on the DNABERT-2 model. Initially, BERT provides a global semantic representation through its contextual extraction capabilities. Subsequently, CNN refines these global features by capturing local patterns within sequences. BiLSTM is then applied to further integrate the features from both modules and to optimize long-range dependency modeling. This integration strategy merges the advantages of global context modeling and local structure extraction, thus forming a comprehensive multi-level feature representation system.

## Promoter prediction using DNABERT-2 variants

The overall architecture of this study is illustrated in Fig. 3. All experiments were conducted on a Windows operating system with Python 3.10 as the programming environment. To accelerate training, an RTX 3090 GPU with 24 GB of VRAM was employed. All models were trained under identical hardware conditions to ensure fair comparisons. The complete source code is available at the following GitHub repository: https://github.com/Cqerliu/DNABERT_CBL.

To investigate the influence of different network architectures on model performance, we designed multiple experimental schemes and proposed two variant strategies aimed at optimizing feature extraction:

Single-module enhancement: DNABERT-2 + CNN and DNABERT-2 + BiLSTM.

This strategy individually augments DNABERT-2 with either a convolutional neural network or a bidirectional long short-term memory network after the BERT encoder. CNN

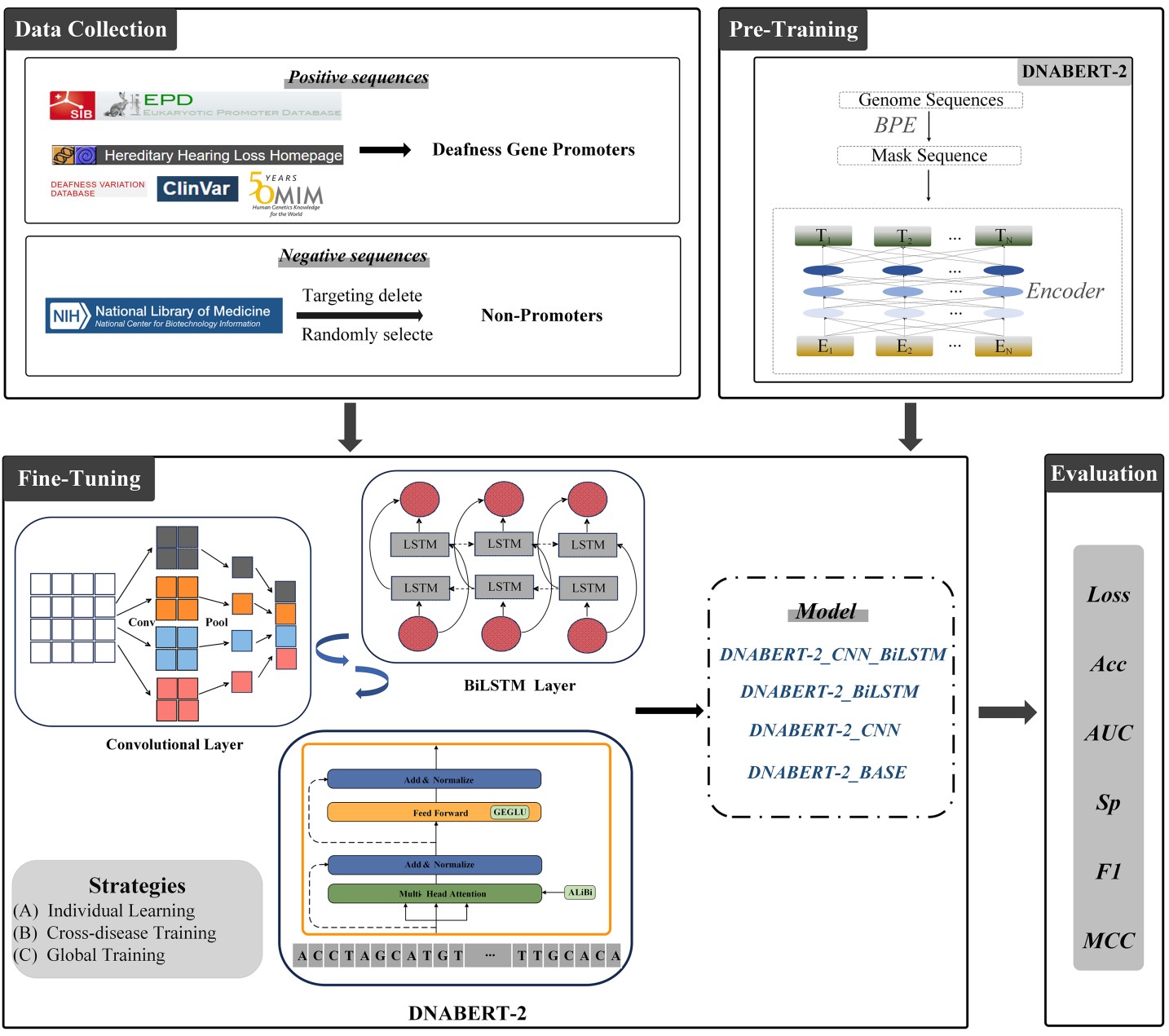

**Figure 3 Prediction of promoters based on enhanced BERT.** The complete workflow of the DNABERT-2 model for predicting gene promoters. The process begins with data collection, where positive sequences (deafness gene promoters) and negative sequences (non-promoters) are sourced from various databases such as ClinVar and the national library of medicine. The model is then fine-tuned using three strategies: individual learning, cross-disease training, and global training. The model's performance is evaluated based on metrics such as loss, accuracy (Acc), AUC, specificity (Sp), F1-score, and MCC. Various model variants are compared to assess their effectiveness in gene prediction tasks.

modules effectively capture local patterns within the input sequences, while BiLSTM modules are well-suited for modeling long-range dependencies. This approach allows us to examine the isolated effects of each module on feature extraction and prediction performance.

Multi-module combination enhancement: DNABERT-2 + CNN + BiLSTM.

This strategy sequentially integrates both CNN and BiLSTM modules to form a comprehensive, multi-module network. Initially, the CNN extracts local features, which are then passed to the BiLSTM layer to model long-distance relationships. This hierarchical design leverages the strengths of both modules, aiming to maximize the overall predictive power.

The specific architectures and hyperparameters for each model are as follows:

(1) Baseline model (DNABERT-2_BASE): This model consists solely of the BERT encoder followed by a fully connected classification layer. The BERT backbone employed is 'zhihan1996/DNABERT_2-117M,' coupled with a single-layer fully connected classifier.

(2) Single-module model (DNABERT-2_CNN): A convolutional layer is added after the BERT output to enhance local feature extraction. The architecture is BERT → convolutional layer → max pooling layer → fully connected classifier. The convolutional layer has a kernel size of 3 and 1,536 output channels, with a pooling window size of 2. The ReLU activation function is used to introduce non-linearity.

(3) Single-module model (DNABERT-2_BiLSTM): A BiLSTM layer is added after the BERT output to capture long-range dependencies. The architecture is BERT → BiLSTM layer → fully connected classifier. The BiLSTM layer has 128 units and uses ReLU activation.

(4) Multi-module model (DNABERT-CBL): This model first applies a CNN layer to extract local features, followed by a BiLSTM layer to model long-distance relationships. The architecture is BERT → convolutional layer → max pooling layer → BiLSTM layer → fully connected classifier. The convolutional layer has a kernel size of 3 and 1,536 output channels, while the BiLSTM layer has 128 units. ReLU activation is consistently applied throughout the model.

All models were trained using the RTX 3090 GPU with 24 GB VRAM to ensure consistent training environments. The hyperparameters used in the experiments are summarized in Table 3. We employed the Adam optimizer with a learning rate of 3e−5 to avoid overfitting and ensure stable convergence. Models were trained for five epochs, with evaluation performed every 200 steps, and the best-performing model on the validation set was saved. A batch size of 8 was used during training to balance GPU memory utilization and computational efficiency, and increased to 16 during evaluation to speed up inference. ReLU activation was chosen throughout for its effectiveness in mitigating gradient vanishing and accelerating convergence. Given the binary classification nature of the task, the cross-entropy loss function was selected for its widespread applicability and proven performance. The total number of parameters for each model is provided in Table 4.

## Evaluation metrics

To comprehensively evaluate the performance of each model, we adopted a suite of well-established classification metrics, including:

Loss: Measures the discrepancy between predicted outputs and true labels, providing insights into convergence and optimization quality. AUC: Assesses the model's capability to distinguish between classes, with higher values indicating better discrimination. ACC:

**Table 3 Network hyperparameters.** Outline of the network hyperparameters used for training the models. It includes key parameters like optimizer type (Adam), learning rate (3e–5), batch sizes for training and evaluation, the number of epochs (5), loss function (crossentropy), and activation function (ReLU).

| Hyperparameter | Value |
|---|---|
| Optimizer | Adam |
| Learning rate | 3e–5 |
| Batch size | 8 (training)/16 (evaluation) |
| Epochs | 5 |
| Loss function | Cross-entropy loss |
| Activation function | ReLU |

**Table 4 Total number of parameters for four models.** Comparison of the total number of parameters for four different model configurations: DNABERT_2, DNABERT_2+CNN, DNABERT_2+BiLSTM, and DNABERT_2+CNN+BiLSTM. The table shows that the model with both CNN and BiLSTM layers (DNABERT_2+CNN+BiLSTM) has the highest parameter count, indicating increased complexity and capacity for learning more intricate patterns from the data.

| Model | Total parameters |
|---|---|
| DNABERT_2 | 117,070,082 |
| DNABERT_2+CNN | 120,615,170 |
| DNABERT_2+BiLSTM | 117,988,610 |
| DNABERT_2+CNN+BiLSTM | 122,318,594 |

The proportion of correctly classified samples among all samples, reflecting overall classification performance. Matthew's correlation coefficient (MCC): A robust metric that accounts for all elements of the confusion matrix, particularly suitable for imbalanced datasets. F1-score: The harmonic mean of precision and recall, balancing the trade-off between false positives and false negatives. Specificity (Sp): Measures the true negative rate, indicating how well the model identifies negative samples.

This combination of metrics offers a comprehensive perspective on each model's strengths and weaknesses, enabling an in-depth analysis of their predictive performance.

## RESULTS

Under the individual learning strategy, five model variants were trained and evaluated: DNABERT_2_BASE, DNABERT_2_CNN, DNABERT_2_BiLSTM, and two DNABERT-CBL models (DNABERT_2_$C_M$_BL and DNABERT_2_$C_A$_BL). Here, "M" and "A" in the subscripts denote the use of max pooling and average pooling, respectively, in the CNN pooling layer. The hearing loss dataset was split into training, validation, and test sets in a 6:2:2 ratio.

The individual learning strategy involved training and testing on the hearing loss dataset, which was divided into training, validation, and test sets in a 6:2:2 ratio. Table 5 summarizes the performance of each model variant in terms of key metrics such as Loss, Accuracy (ACC), and AUC. Notably, DNABERT-2_$C_A$_BL achieved the best performance in both 300 and 600 bp settings, demonstrating superior classification capabilities

**Table 5 Performance comparison of various models with different sequence lengths under individual learning strategies.** A performance comparison of various models using different sequence lengths (300 and 600 bp) under individual learning strategies. The metrics include loss, accuracy (Acc), AUC, F1-score, specificity (Sp), and Matthews correlation coefficient (MCC). The results show how models like DNABERT-2_BASE, DNABERT-2_CNN, and DNABERT-2_BiLSTM perform differently across these sequence lengths. The best results for sequence lengths of 300 bp and 600 bp are indicated in bold.

| Length | Model | Loss | Acc | AUC | F1 | Sp | MCC |
|---|---|---|---|---|---|---|---|
| 300 bp | DNABERT-2_BASE | 0.518 | 0.850 | 0.895 | 0.849 | 0.933 | 0.710 |
| | DNABERT-2_CNN | 0.509 | 0.858 | **0.931** | 0.858 | 0.925 | 0.723 |
| | DNABERT-2_BiLSTM | 0.467 | **0.867** | 0.841 | 0.865 | **0.967** | **0.749** |
| | **DNABERT-2_C$_M$_BL** | **0.395** | 0.858 | 0.924 | 0.857 | 0.942 | 0.727 |
| | **DNABERT-2_C$_A$_BL** | 0.495 | **0.867** | 0.903 | **0.866** | 0.942 | 0.742 |
| 600 bp | DNABERT-2_BASE | 0.537 | 0.846 | 0.916 | 0.843 | **0.975** | 0.716 |
| | DNABERT-2_CNN | 0.567 | 0.871 | 0.940 | 0.870 | 0.958 | 0.753 |
| | DNABERT-2_BiLSTM | 0.516 | 0.867 | 0.909 | 0.867 | 0.883 | 0.734 |
| | **DNABERT-2_C$_M$_BL** | 0.429 | 0.842 | 0.931 | 0.839 | 0.958 | 0.703 |
| | **DNABERT-2_C$_A$_BL** | **0.333** | **0.904** | **0.949** | **0.904** | 0.933 | **0.810** |

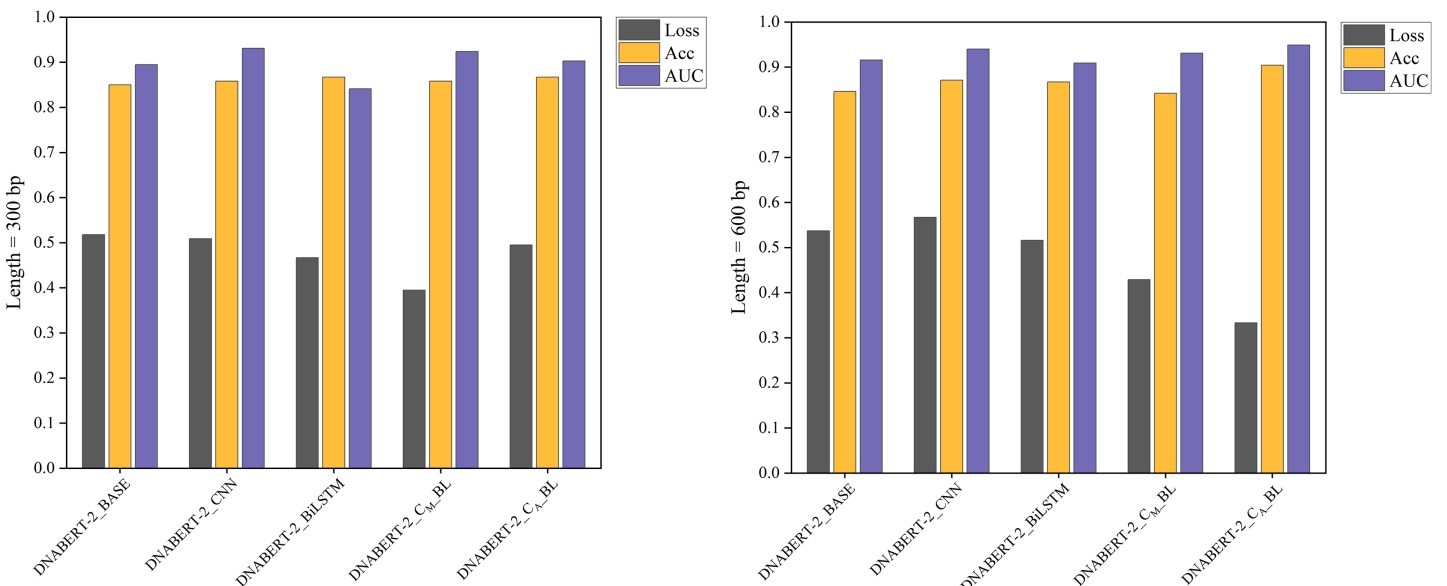

**Figure 4 Performance comparison of five models on loss, accuracy, and AUC metrics (individual learning).** Comparison of the performance of five different models based on three metrics: loss, accuracy, and AUC. Each model's performance is assessed after individual learning, using genome sequences of varying lengths (300 and 600 bp). The results show the models' effectiveness in prediction, with the DNABERT-CBL model generally outperforming others in terms of accuracy and AUC.                               

compared to the baseline. These results highlight the differences in performance across models under varying sequence lengths, with comparative results illustrated in Fig. 4.

With the individual learning strategy, DNABERT-CBL performed exceptionally well on both 300 and 600 bp sequence lengths. Specifically, DNABERT-CBL with average pooling achieved a loss of 0.333 and an AUC of 0.949 for 600 bp, thus significantly outperforming

**Table 6 Performance comparison of models with different sequence lengths on the breast cancer dataset.** Comparison of the performance of models with different sequence lengths (300 and 600 bp) on the breast cancer dataset. Key performance metrics, including loss, accuracy, AUC, F1-score, specificity, and MCC, are displayed. It shows that models like DNABERT-2_CA_BL perform exceptionally well, particularly with the 600 bp sequence length, achieving high accuracy and AUC scores. The best results for sequence lengths of 300 bp and 600 bp are indicated in bold.

| Length | Model | Loss | Acc | AUC | F1 | Sp | MCC |
|--------|-------|------|-----|-----|-----|-----|-----|
| 300 bp | DNABERT-2_BASE | 0.475 | 0.851 | 0.930 | 0.849 | **0.975** | 0.724 |
|  | DNABERT-2_CNN | 0.364 | 0.855 | 0.921 | 0.854 | 0.928 | 0.718 |
|  | DNABERT-2_BiLSTM | 0.561 | **0.878** | 0.920 | **0.878** | 0.883 | 0.755 |
|  | **DNABERT-2_C$_M$_BL** | 0.410 | 0.853 | 0.932 | 0.850 | **0.975** | 0.727 |
|  | **DNABERT-2_C$_A$_BL** | **0.301** | 0.876 | **0.955** | 0.876 | 0.941 | **0.759** |
| 600 bp | DNABERT-2_BASE | 0.363 | 0.884 | 0.933 | 0.883 | 0.913 | 0.768 |
|  | DNABERT-2_CNN | 0.481 | **0.890** | 0.933 | **0.890** | 0.955 | **0.787** |
|  | DNABERT-2_BiLSTM | 0.650 | 0.775 | 0.906 | 0.764 | **0.988** | 0.608 |
|  | **DNABERT-2_C$_M$_BL** | **0.316** | 0.874 | 0.950 | 0.874 | 0.956 | 0.759 |
|  | **DNABERT-2_C$_A$_BL** | 0.324 | 0.875 | **0.953** | 0.874 | 0.963 | 0.762 |

the baseline DNABERT-2_BASE, which had a loss of 0.537 and an AUC of 0.916. Figure 4 highlights DNABERT-CBL's superior performance across key metrics.

## Results of the cross-disease training strategy

In the cross-disease training strategy, the models were pre-trained on specific disease datasets (breast cancer and cervical cancer) before testing on the hearing loss dataset. For the breast cancer dataset, training and validation sets were split 8:2, followed by testing on the hearing loss dataset. The cervical cancer dataset was similarly processed. Tables 6 and 7, along with Figs. 5 and 6, display the performance metrics and comparisons. In both cases, DNABERT-2_C$_A$_BL consistently achieved the highest ACC and F1-scores, demonstrating strong generalization ability from breast and cervical cancer datasets to hearing loss promoters.

In cross-disease training, DNABERT-CBL also excelled. For example, with training on the breast cancer dataset, the variant with average pooling achieved an AUC of 0.955, compared to the baseline's 0.930. Similarly, for the cervical cancer dataset, DNABERT-CBL achieved an AUC of 0.948 at 600 bp, surpassing the baseline's 0.931. Figures 5 and 6 further validate the model's advantages in transfer learning.

## Results of the global training strategy

The global training strategy involved training on a dataset containing all known human promoter sequences, excluding hearing-loss-related data, and then testing on the hearing loss dataset. Promoter sequences associated with hearing loss were removed from the training data, which was then split 8:2 for training and validation. Table 8 provides the performance metrics, and Fig. 7 visually compares the models' performance, showing that DNABERT-CBL consistently outperformed the baseline model in terms of stability and AUC across both sequence lengths.

**Table 7 Transfer learning performance comparison of models with different sequence lengths on the cervical cancer dataset.** The transfer learning performance of different models using 300 and 600 bp sequence lengths on the cervical cancer dataset. The table includes various evaluation metrics such as loss, accuracy, AUC, F1-score, specificity, and MCC. The results suggest that DNABERT-2_CA_BL performs the best across most metrics, especially with 600 bp sequences. The best results for sequence lengths of 300 bp and 600 bp are indicated in bold.

| Length | Model | Loss | Acc | AUC | F1 | Sp | MCC |
|--------|-------|------|-----|-----|----|----|----|
| 300 bp | DNABERT-2_BASE | 0.313 | 0.872 | 0.931 | 0.872 | 0.827 | 0.747 |
| | DNABERT-2_CNN | 0.372 | 0.873 | 0.933 | 0.873 | 0.891 | 0.746 |
| | DNABERT-2_BiLSTM | 0.379 | 0.884 | 0.874 | 0.884 | **0.950** | 0.775 |
| | **DNABERT-2_$C_M$_BL** | 0.309 | **0.899** | 0.942 | **0.899** | 0.931 | **0.799** |
| | **DNABERT-2_$C_A$_BL** | **0.290** | 0.894 | **0.949** | 0.894 | 0.901 | 0.789 |
| 600 bp | DNABERT-2_BASE | 0.362 | 0.885 | 0.931 | 0.885 | 0.868 | 0.771 |
| | DNABERT-2_CNN | 0.472 | 0.889 | 0.939 | 0.889 | 0.868 | 0.778 |
| | DNABERT-2_BiLSTM | 0.470 | 0.882 | 0.900 | 0.882 | 0.881 | 0.764 |
| | **DNABERT-2_$C_M$_BL** | 0.315 | **0.895** | **0.949** | **0.895** | **0.908** | **0.791** |
| | **DNABERT-2_$C_A$_BL** | **0.310** | 0.893 | 0.948 | 0.893 | 0.881 | 0.786 |

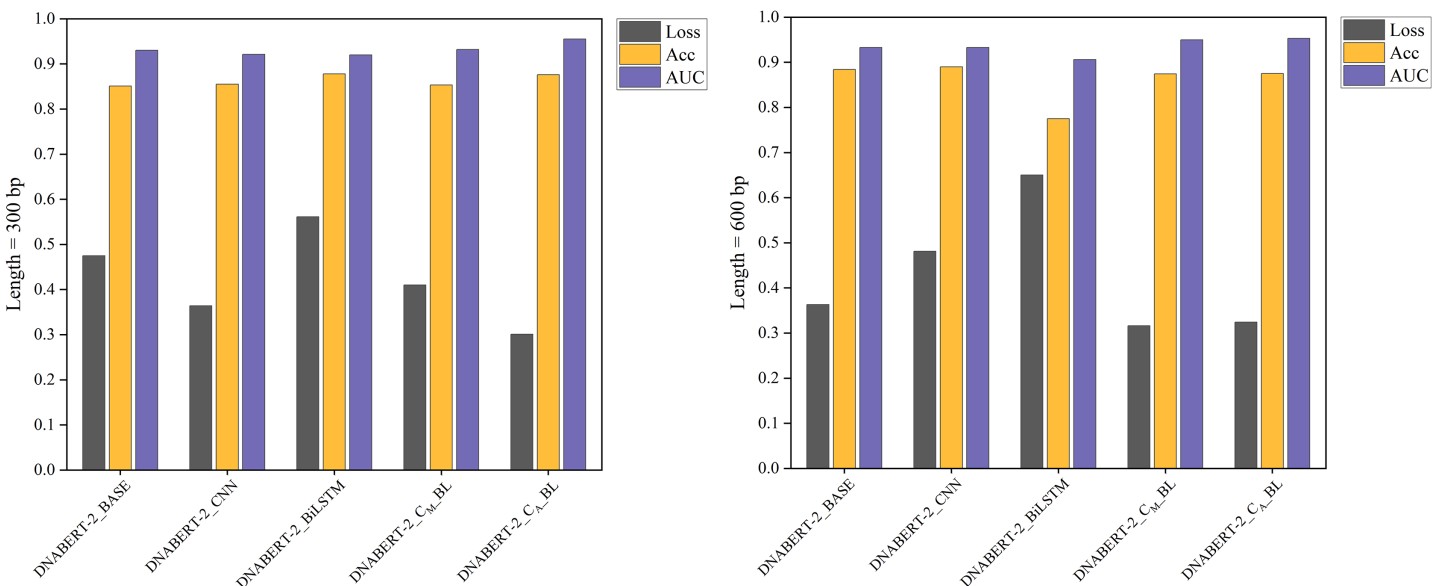

**Figure 5 Performance comparison of five models on loss, accuracy, and AUC metrics (breast cancer dataset).** A comparison of five models' performance on the breast cancer dataset. The chart highlights the models' loss, accuracy, and AUC metrics, showcasing their predictive capabilities for this specific dataset. The DNABERT-CBL model consistently delivers high performance, particularly in accuracy and AUC.

Under the global training strategy, DNABERT-CBL again demonstrated strong performance, consistently achieving lower loss and higher accuracy across both sequence lengths. It also exhibited greater AUC stability, as shown in Fig. 7.

## Comprehensive evaluation

Across all strategies, DNABERT-CBL demonstrated superior performance, particularly on large datasets and complex tasks. The AUC heatmap in Fig. 8 further illustrates its

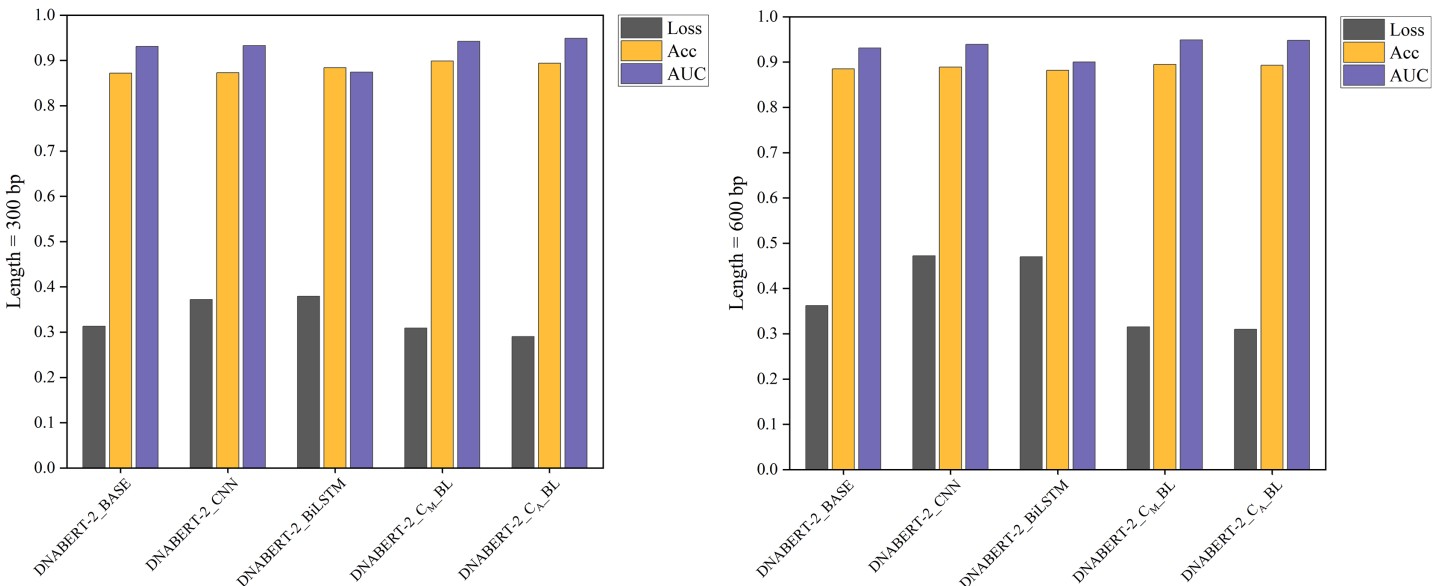

**Figure 6 Performance comparison of five models on loss, accuracy, and AUC metrics (cervical cancer dataset).** A comparison of the performance of five different models on the cervical cancer dataset using loss, accuracy, and AUC metrics. The DNABERT-CBL model demonstrates strong performance, particularly in terms of accuracy and AUC, making it a robust choice for predicting cancer-related genomic features.

**Table 8 Performance comparison of models with different sequence lengths under global training.** The performance comparison of models using different sequence lengths (300 and 600 bp) under global training. It provides insights into the effectiveness of each model based on loss, accuracy, AUC, F1-score, specificity, and MCC. The DNABERT-2_CM_BL model performs exceptionally well under this strategy, particularly with 300 bp sequences. The best results for sequence lengths of 300 bp and 600 bp are indicated in bold.

| Length | Model | Loss | Acc | AUC | F1 | Sp | MCC |
|--------|-------|------|-----|-----|-----|-----|-----|
| 300 bp | DNABERT-2_BASE | 0.258 | 0.894 | 0.964 | 0.894 | 0.883 | 0.789 |
| | DNABERT-2_CNN | **0.232** | 0.907 | 0.969 | 0.907 | 0.910 | 0.814 |
| | DNABERT-2_BiLSTM | 0.296 | 0.898 | 0.940 | 0.898 | 0.905 | 0.796 |
| | **DNABERT-2_C$_M$_BL** | 0.243 | **0.909** | **0.970** | **0.909** | 0.923 | **0.818** |
| | **DNABERT-2_C$_A$_BL** | 0.242 | 0.905 | 0.965 | 0.904 | **0.935** | 0.811 |
| 600 bp | DNABERT-2_BASE | 0.257 | 0.905 | 0.963 | 0.905 | 0.891 | 0.811 |
| | DNABERT-2_CNN | **0.239** | 0.910 | 0.966 | 0.910 | 0.905 | 0.819 |
| | DNABERT-2_BiLSTM | 0.275 | 0.911 | 0.950 | **0.911** | **0.913** | **0.822** |
| | **DNABERT-2_C$_M$_BL** | 0.255 | 0.910 | **0.970** | 0.910 | 0.908 | 0.819 |
| | **DNABERT-2_C$_A$_BL** | 0.273 | **0.904** | 0.962 | 0.904 | 0.889 | 0.808 |

consistent advantage. DNABERT-CBL excelled in loss, accuracy, and AUC, outperforming the baseline and exhibiting robust adaptability to diverse sequence lengths and datasets.

In summary, the DNABERT-CBL model demonstrates outstanding performance in both individual learning and transfer learning, making it especially suitable for handling sequences of varying lengths and complex datasets. It exhibits strong adaptability and robustness across different conditions.

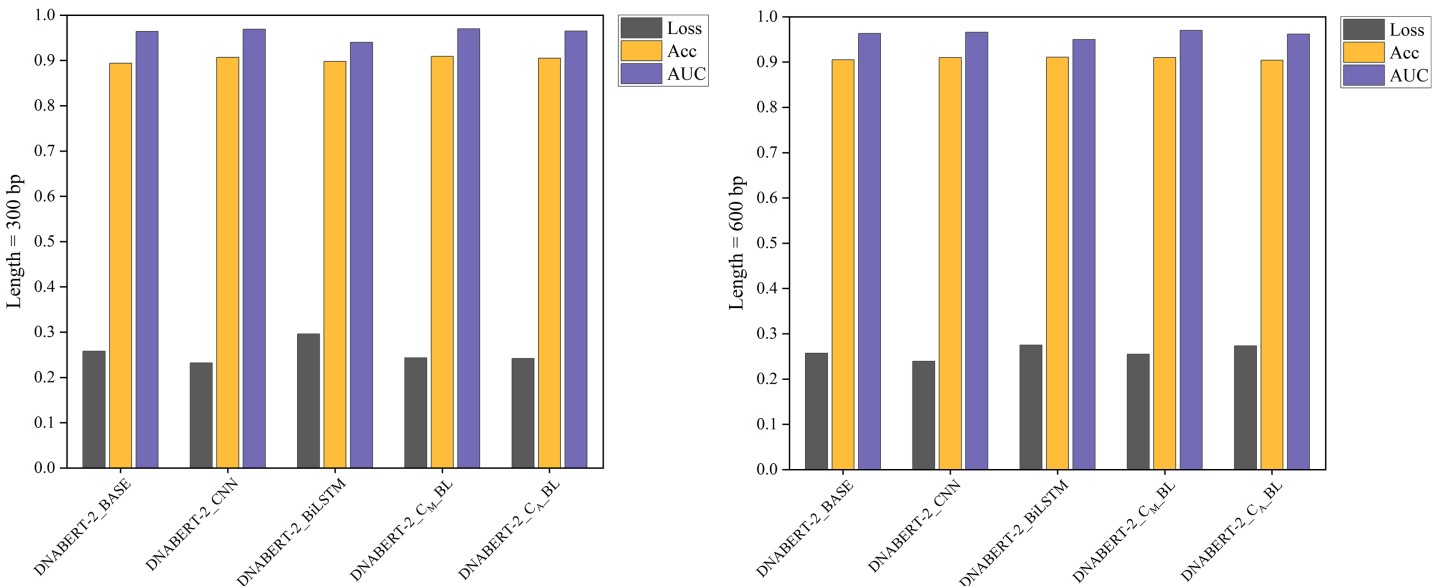

**Figure 7 Performance comparison of five models on loss, accuracy, and AUC metrics (global training).** The performance comparison of five models on global training, evaluating them based on loss, accuracy, and AUC metrics. The DNABERT-CBL model continues to excel across all metrics, reinforcing its effectiveness when applied globally across different datasets.

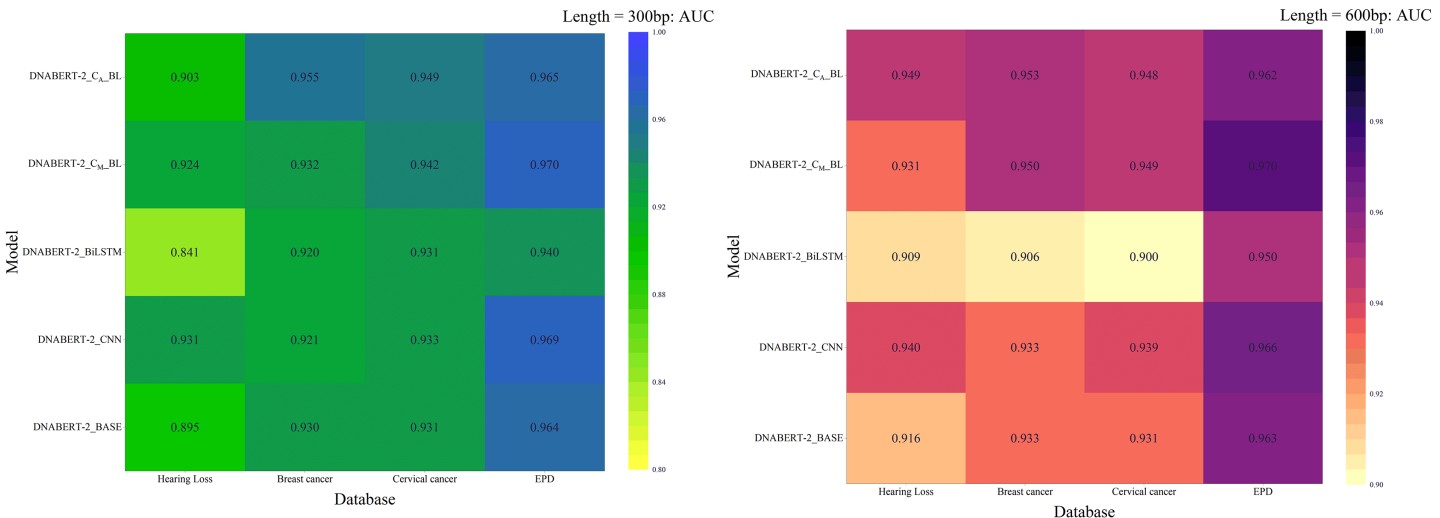

**Figure 8 AUC heat maps of the model under three strategies.** AUC scores of the DNABERT-CBL model under three different training strategies (individual learning, cross-disease training, and global training). Each map highlights the AUC performance across multiple datasets, with higher scores represented in warmer colors, indicating that the model performs better with certain strategies, especially in global training scenarios.

## Comparison with other BERT-based methods for predicting promoters

Currently, few studies utilize BERT and its variants for promoter prediction. To evaluate our model's effectiveness, we compared it with three state-of-the-art BERT-based methods: BERT-Promoter (*Le et al., 2022*), msBERT-Promoter (*Li et al., 2024*), and TSSNote-CyaPromBERT (*Mai, Nguyen & Lee, 2022*). All models were tested on the same

**Table 9 Performance comparison between this method and other BERT-based methods.** Comparison the performance of DNABERT-CBL with other BERT-based models on key metrics such as accuracy (Acc), F1-score, MCC, precision, sensitivity (Sn), and AUC. DNABERT-2_CA_BL outperforms other models like BERT-Promoter and msBERT-Promoter across multiple metrics, achieving high accuracy, F1, and AUC scores. The maximum value of each column is in bold.

| Model | Acc | F1 | MCC | Precision | Sn | AUC |
|---|---|---|---|---|---|---|
| **DNABERT-2_$C_M$_BL** | 0.858 | 0.857 | 0.727 | 0.869 | 0.858 | 0.924 |
| **DNABERT-2_$C_A$_BL** | **0.867** | **0.866** | **0.742** | **0.875** | **0.867** | 0.903 |
| BERT-Promoter | 0.848 | 0.846 | – | 0.864 | 0.829 | 0.918 |
| msBERT-Promoter | 0.836 | 0.833 | 0.696 | 0.861 | 0.836 | **0.939** |
| TSSNote-CyaPromBERT | 0.822 | 0.822 | – | 0.824 | 0.820 | 0.899 |

benchmark dataset, splitting the hearing loss promoter sequences into training, validation, and test sets at a 6:2:2 ratio. Due to the 512-bp input length limitation of these models, only 300 bp experiments were conducted. Table 9 presents the performance metrics across all methods.

Our models, DNABERT-2_$C_M$_BL and DNABERT-2_$C_A$_BL, achieved the highest scores in accuracy, F1-score, Matthew's correlation coefficient (MCC), and precision. Notably, DNABERT-2_$C_A$_BL with average pooling achieved an accuracy of 0.867, an F1-score of 0.866, an MCC of 0.742, and a precision of 0.875, demonstrating robust classification capability. Although its AUC (0.903) was slightly lower than that of msBERT-Promoter (0.939), this discrepancy may be attributed to a relatively weaker ability to discriminate borderline samples, thereby slightly reducing the area under the ROC curve. However, considering that AUC reflects an overall ranking performance, while DNABERT-2_$C_A$_BL achieved higher scores in core classification metrics such as accuracy and F1-score, this suggests that our model is more reliable and practically effective in binary classification tasks. Overall, the performance of DNABERT-2_$C_A$_BL still outperformed existing methods, affirming the model's effectiveness and stability in promoter prediction.

## DISCUSSION

Gene sequences encompass a wealth of local regulatory sites (*He et al., 2010*), due to their bidirectional, short-range patterns, and interactions with distal regulatory elements. As a key region in gene transcription, promoters are central to gene regulation and expression, interacting with distal regulatory elements through the 3D chromatin structure. This poses a significant challenge for models: the ability to capture both local features and global dependencies within promoter sequences. The BERT self-attention mechanism alone struggles with such complex regulation patterns. We developed a multi-level feature extraction model, DNABERT-CBL, integrating BERT, CNN, and BiLSTM to address these complexities. The CNN module captures local regulatory sites and short-range patterns within promoter sequences through convolutional operations, while BiLSTM captures long-range dependencies, compensating for BERT's limitations in modeling extended dependencies. This hierarchical feature extraction approach enables the model to globally understand sequence structures while accurately identifying local features.

Experimental results confirmed DNABERT-CBL's effectiveness, as it consistently outperformed the baseline model DNABERT-2_BASE across multiple metrics. For example, in testing with longer sequences, DNABERT-CBL demonstrated significant reductions in loss and improvements in accuracy and AUC, achieving a loss of 0.333, an accuracy of 0.904, and an AUC of 0.949 for 600-bp sequences, compared to the baseline's loss of 0.537 and AUC of 0.916. This highlights the model's strong predictive capabilities for complex gene sequences, with its stability and generalization validated across multiple datasets, particularly on larger and more complex datasets.

## CONCLUSIONS

Promoters are crucial for initiating gene transcription, playing a key role in gene regulation and expression. Mutations in promoters are linked to complex diseases such as diabetes, cancer, and deafness. Variations in promoter sequences also present potential targets for gene therapy. For example, promoter-driven gene therapies, such as Myo15 promoter-mediated hair cell-specific gene therapy, show promising potential in treating autosomal recessive deafness. Future research could leverage the DNABERT-CBL model to further explore the role of promoters in gene expression regulation, particularly in developing gene therapies for complex diseases like deafness. This model's ability to accurately identify mutated promoters could pave the way for new therapeutic strategies targeting gene regulation-related diseases.

The DNABERT-CBL model was chosen for its effectiveness in addressing the complexities of gene sequence analysis. The combination of BiLSTM and CNN enables the model to capture both sequential dependencies and local features in promoter sequences, thereby enhancing prediction accuracy. This multi-layered feature extraction approach makes the model particularly well-suited for understanding gene regulation at the sequence level. By focusing on relevant promoter regions, the model can provide meaningful insights into gene expression regulation.

However, since the current model is based only on gene sequence data, it may not be able to comprehensively capture a wider range of regulatory factors that affect gene expression. Future studies may consider introducing multi-omics data such as epigenomics and transcriptomics to improve the comprehensiveness and predictive power of the model. In addition, the model should be tested on a wider range of promoter sequences and additional data sources should be integrated to further optimize its performance. By addressing these challenges, the model is expected to significantly enhance gene therapy strategies for diseases associated with promoter mutations. Overall, the model shows great potential in advancing promoter-driven gene therapy and provides a valuable tool for gene regulation research and therapeutic development.

## ACKNOWLEDGEMENTS

The authors utilized generative AI tools (ChatGPT 4.5) solely for English language editing and proofreading. The AI tools were not involved in the study design, data analysis, or interpretation of results.

### Funding

This work was supported by the Natural Science Foundation of Chongqing (No. CSTB2024NSCQ-MSX0129). The funders had no role in study design, data collection and analysis, decision to publish, or preparation of the manuscript.

### Grant Disclosures

The following grant information was disclosed by the authors:
Natural Science Foundation of Chongqing: CSTB2024NSCQ-MSX0129.

### Competing Interests

The authors declare that they have no competing interests.

### Author Contributions

- Jing Sun conceived and designed the experiments, performed the experiments, analyzed the data, performed the computation work, prepared figures and/or tables, authored or reviewed drafts of the article, and approved the final draft.
- Yangfan Huang analyzed the data, prepared figures and/or tables, and approved the final draft.
- Jiale Fu performed the computation work, prepared figures and/or tables, and approved the final draft.
- Li Teng conceived and designed the experiments, prepared figures and/or tables, and approved the final draft.
- Xiao Liu conceived and designed the experiments, authored or reviewed drafts of the article, and approved the final draft.
- Xiaohua Luo analyzed the data, prepared figures and/or tables, and approved the final draft.

### Data Availability

Code and raw data is available at GitHub: https://github.com/Cqerliu/DNABERT_CBL.

The ClinVar dataset is available at the National Center for Biotechnology Information (NCBI): https://www.ncbi.nlm.nih.gov/clinvar/?term=deafness.

The Deafness Variation Database (DVD) dataset is available at Molecular Otolaryngology and Renal Research Laboratories (MORL), University of Iowa: https://deafnessvariationdatabase.org/download.

The Online Mendelian Inheritance in Man (OMIM) is available at Johns Hopkins University/National Center for Biotechnology Information (NCBI): https://www.omim.org/search?index=entry&search=deafness&start=1&limit=10&retrieve=geneMap&genemap_exists=true.

**This information can also be found** through the OMIM website (https://www.omim.org/) by entering the keyword "deafness" in the search box on the homepage and then selecting the "Gene Map" option in the retrieval filter.

The Hereditary Hearing Loss (HHL) dataset is available at Radboud University Medical Center:

- https://hereditaryhearingloss.org/syndromic.
- https://hereditaryhearingloss.org/nonsyndromic.

The METABRIC dataset is available at the European Genome-phenome Archive (EGA): https://ega-archive.org/dacs/EGAC00001000484.

The Cervical Cancer Database (CCDB) is available at the Bioinformatics Centre, Institute of Microbial Technology: http://crdd.osdd.net/raghava/ccdb/stat.php.

The EPDnew dataset is available at the Swiss Institute of Bioinformatics: https://epd.expasy.org/epd/human/human_database.php?db=human.

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
