# Peer review of "An enhanced BERT model with improved local feature extraction and long-range dependency capture in promoter prediction for hearing loss"

_PeerJ Computer Science, doi:10.7717/peerj-cs.3104_

## Round 0.1 · original submission · Major Revisions

Please follow the reviewers' requests and comments in detail.

**Language Note:** The review process has identified that the English language must be improved. PeerJ can provide language editing services - please contact us at [email protected] for pricing (be sure to provide your manuscript number and title). Alternatively, you should make your own arrangements to improve the language quality and provide details in your response letter. – PeerJ Staff

Reviewer 1 ·

Basic reporting

The paper proposes a deep learning model with DNABERT-2 combined with CNN and Bi-LSTM for promoter prediction in hearing loss. Although the premise is interesting, there are several flaws in the manuscript.
1) The Introduction is clearly written to show the context. However, going forward in the manuscript, it is difficult to understand why the authors have mentioned hearing loss in the title when clearly there is no importance or emphasis given to this particular point in the later part of the manuscript. I suggest revising the title of the paper.
2) Please keep a space between the text and the citations.

Experimental design

The authors have provided all the code and data used in this paper as a GitHub repository. However, in data preprocessing, some parts are not clear.
1) The authors have mentioned in lines 140-142 of the article "To assess the impact of sequence length on the experimental results, we conducted studies with two sequence lengths: 300 bp[29]([-249, +50]) and 600 bp[30]([-499, +100])". For a general reader, understanding +50 and +100 may prove difficult. Please elaborate.
2) The authors have designed three experiments like individual training, cross-disease training and global training data. Cross-disease is based on training conducted on the breast cancer and cervical cancer datasets, with testing on the hearing loss dataset. It is difficult to understand what is the relation between the cancer and the hearing loss dataset and what kind of knowledge are we gaining from this experiment.
3) The dataset is too small to conclude anything meaningful. Is there any specific reason for such small dataset?
4) The hyperparameters for CNN and Bi-LSTM are not mentioned,
5) What are DNABERT-2_CM_BL and DNABERT-2_CA_BL?

Validity of the findings

The authors have provided all the codes and datasets, and they have also done proper ablation studies. However, the dataset is too small. Also, there are works like "iProL: identifying DNA promoters from sequence information based on Longformer pre-trained model" that have used CNN and Bi-LSTM with DNA embeddings. It is important to point out the contributions and the novelty of the work.

Reviewer 2 ·

Basic reporting

Abstract—The abstract is poorly written and needs updating, as it lacks clarity of the overall work. Suggest rewriting the abstract and not using Gen AI adopted abstract.

The figures and tables are poorly described within the manuscript and the captions needs updating.
- Minor typographical and grammatical issues exist throughout (e.g., line 17: “genetic data's unique features4local regulatory sites” has encoding errors).
-The naming convention of the models (e.g., DNABERT-2_CM_BL, DNABERT-2_CA_BL) is not intuitive and should be clarified earlier (perhaps with a key in Table 3 or 7).
-Figures 1–3 are referenced, but their clarity could be improved with more detailed legends and possibly architecture-layer diagrams.
- All the Tables and captions needs updating in detail to give clarity.

Experimental design

Experimental Design:
The good part is the codebase is publicly available, which supports reproducibility: https://github.com/Cqerliu/DNABERT-2_CNN_BiLSTM.
Please see below comments to improve.

-The model was trained only on sequence data, excluding epigenomic features like methylation or chromatin accessibility, which could improve promoter prediction especially in hearing-related tissues.
-Computing infrastructure lacks detail. It is only stated that the experiments used RTX 3090 with 24GB RAM. Batch sizes, training epochs, learning rates, and dropout rates are not fully specified.
-No statistical tests are used to demonstrate whether performance differences between models are significant. Confidence intervals or p-values for AUC, ACC, or F1 would strengthen claims.

Validity of the findings

Validity of the finding

Strengths:

- The model performance improvements are well-demonstrated with consistent gains across datasets and strategies.
- The model consistently outperforms not only the DNABERT-2 baseline but also other state-of-the-art promoter models like BERT-Promoter and msBERT-Promoter (Table 7).
- Transfer learning from breast and cervical cancer datasets demonstrates good generalisation capacity for the hearing loss dataset, reinforcing cross-disease utility.


Weakness:

- The dataset is moderately sized (1,099 genes), and while training on 59,195 promoter sequences in global training is impressive, generalisation to under-represented classes or promoters from novel conditions is not tested.
- There is limited discussion on why AUC for DNABERT-2_CA_BL is slightly lower than msBERT-Promoter in Table 7 despite better performance on other metrics.

Additional comments

The paper is reasonably written with technical detail. The hybrid architecture addresses known limitations in BERT's handling of local vs. long-range dependencies in genomic sequences. Its biological focus on hearing loss is novel and timely, given the emergence of gene therapies targeting auditory function.

Given the good results and solid reproducibility (via GitHub), I recommend revisions to update and then a decision can be taken. The key issues are mostly polish-related (statistical support, better naming, and transparency in experimental design).

---

## Round 0.2 · accepted · Accept

Thank you for providing the requested changes to the manuscript.

Reviewer 2 ·

Basic reporting

The authors have accepted the suggestions and worked on improving the paper as required

Experimental design

The authors have accepted the suggestions and worked on improving the paper as required

Validity of the findings

Thanks for accepting the suggestions to improve the validation method